# Artificial Reproduction of Blue Bream (*Ballerus ballerus* L.) as a Conservative Method under Controlled Conditions

**DOI:** 10.3390/ani11051326

**Published:** 2021-05-06

**Authors:** Przemysław Piech, Roman Kujawa

**Affiliations:** Department of Ichthyology and Aquaculture, Faculty of Animal Bioengineering, University of Warmia and Mazury in Olsztyn, PL 10-719 Olsztyn, Poland; reofish@uwm.edu.pl

**Keywords:** rheophilic fish, controlled conditions, artificial reproduction, hormonal stimulation

## Abstract

**Simple Summary:**

Quite severe biological imbalances have been caused by the often ill-conceived and destructive actions of humans. The natural environment, with its flora and fauna, has been subjected to a strong, direct or indirect, anthropogenic impact. In consequence, the total population of wild animals has been considerably reduced, despite efforts to compensate for these errors and expand the scope of animal legal protection to include endangered species. Many animal populations on the verge of extinction have been saved. These actions are ongoing and embrace endangered species as well as those which may be threatened with extinction in the near future as a result of climate change. The changes affect economically valuable species and those of low value, whose populations are still relatively strong and stable. Pre-emptive protective actions and developing methods for the reproduction and rearing of rare species may ensure their survival when the ecological balance is upset. The blue bream is one such species which should be protected while there is still time.

**Abstract:**

The blue bream *Ballerus ballerus* (L.) is one of two species of the *Ballerus* genus occurring in Europe. The biotechnology for its reproduction under controlled conditions needs to be developed to conserve its local populations. Therefore, a range of experiments were performed to obtain valuable gametes from blue bream following hormonal stimulation with carp pituitary homogenate (CPH), Ovopel and Ovaprim. CPH and Ovopel were injected twice and Ovaprim—once, under a pectoral fin. The spawners were inspected 12 h after the last injection, and the gametes were collected. Eggs were collected from spawning females and fertilized with sperm from three males. The smallest number of ovulating blue bream (four females) was obtained from individuals stimulated with Ovaprim. There were six to nine ovulating females in the other groups, accounting for 60–90% of the females in the respective groups. The latency period was the shortest in the spawners stimulated with Ovopel. The fish in this group ovulated 14 h after the hormonal injection. Spawning was synchronized and all females spawned simultaneously. The highest average weight of eggs per fish (75.4 g) was obtained from the females stimulated with Ovopel, whereas the individuals stimulated with Ovaprim provided the smallest average amount of eggs (49.5 g). The study showed that blue bream spawners are not very sensitive to reproduction-related handling under controlled conditions.

## 1. Introduction

There are many fish species of high or low economic value inhabiting inland waters in Poland. Fish of both types are often caught together. Although the economic importance of some fish is low, they are an essential component of the ichthyofauna, affecting the aquatic environment biodiversity. One such species is the blue bream *Ballerus ballerus* (L.)—one of two representatives of the genus *Ballerus* occurring in Europe [1]. It is a phytophilous rheophilic cyprinid fish, inhabiting large, slow lowland rivers, flow-through swamps and eutrophic lakes with lush vegetation. The blue bream mainly inhabits the pelagic zone in lakes and the benthopelagic zone in rivers [2]. It is often found in salty and brackish waters in river estuaries of the North and Baltic Seas, and of the European parts of the Black Sea, the Sea of Azov and the Caspian Sea river basins [3].

The fish reach sexual maturity at the age of 3–5 years [4]. Mature females are at least 24.1 cm long and mature males are 20.6 cm in length [5,6]. This fish produces eggs once a season, usually between early April and early May, in water above 12 °C [6,7]. The spawning period can shift to late May if the water does not become warm enough due to low spring air temperatures [5]. The blue bream’s annual reproductive cycle description is based on gonad morphology, GSI fluctuations [5] and the gonad histology in fish of both sexes in the annual development cycle [8,9].

This species’ reproduction under controlled conditions has not yet been conducted. The issue is very important given the fact that the procedure would help to maintain or restore its population should such a need arise in the future. Unlike other cyprinids, the blue bream is sensitive to thermal water pollution. The forecast water temperature increase due to climate change could have a considerable impact on the species’ range in the wild [10]. The water temperature has a great impact on the seasonal spawning effectiveness and, consequently, on the condition of ichthyofauna in inland waters [11]. Climate change and extreme weather conditions may upset natural fish reproduction [12]. Disruption of natural reproductive functions may, in extreme situations, lead to the total disappearance of individual fish species populations. Developing methods for reproduction under controlled conditions for fish species of low economic value and larvae rearing in recirculation aquaculture systems may considerably boost these species’ stocking material production. Production intensification in Recirculating Aquaculture Systems (RAS) provides a great opportunity for the development of conservative aquaculture. The research conducted thus far has facilitated the implementation of the developed technologies in fishing practice. This has increased stocking with such biologically valuable species as the asp *Leuciscus aspius*, the ide *Leuciscus idus*, the common rudd *Scardinius erythrophthalmus*, the dace *Leuciscus leuciscus*, the common bream *Abramis brama* and the burbot *Lota lota* [12,13,14,15,16,17,18,19,20]. Maintaining the population sizes of low-economic value fish at an appropriate level ensures sufficient feed for such fish as the pike *Esox lucius* and the European catfish *Silurus glanis*. Stocking with fish species of low economic value also increases the biodiversity in water bodies and watercourses.

Fish reproduction under controlled conditions is a common method for supporting fish populations. There are many methods for aiding fish reproduction. Oogenesis and spermatogenesis are induced with chemical preparations. Without hormonal stimulation, it is impossible to obtain mature oocytes from female fish of most cyprinid species under controlled conditions [17,21,22]. Hormonal agent application to male fish increases the amount of sperm obtained from them and improves its quality [23]. Hormonal stimulation combined with a proper thermal regime is the most effective method. Maintaining the appropriate temperature together with hormone application is a guarantee of obtaining valuable gametes [24]. The thermal requirements are adjusted on a species-specific basis. Maintaining the assumed temperature-related regime has a huge impact on the development of maturing oocytes and successful spawning. Even small temperature fluctuations often lead to gonad resorption [11]. The method for fish reproduction under the controlled conditions described above is usually applied together with an appropriate photoperiod [25,26,27,28,29].

The professional literature provides descriptions of many hormonal agents with a beneficial impact on fish reproduction. The effectiveness of these agents varies. The most common agents administered to fish include carp pituitary homogenate (CPH), human chorionic gonadotropin (hCG), Ovopel containing a mammalian gonadotropin-releasing hormone (GnRH) analogue and metoclopramide as a dopamine receptor antagonist and Ovaprim containing an analogue of salmon GnRH and domperidone as a dopamine receptor antagonist [12,17,30,31]. Most hormonal agents require proper preparation—they must be dissolved or homogenized in a 0.9% NaCl solution [32,33,34]. Among the hormonal agents mentioned above, only Ovaprim is immediately ready for use. The other reproduction stimulators must be homogenized and dissolved in physiological saline. The injections are given intraperitoneally or intramuscularly [27].

In this study, blue bream reproduction under controlled conditions was performed following stimulation with the three preparations most commonly used in cyprinid fish. The study aimed to develop a procedure for blue bream reproduction under controlled conditions.

## 2. Materials and Methods

The blue bream reproductive study was conducted at the Centre for Aquaculture and Ecological Engineering in the Department of Ichthyology and Aquaculture of the University of Warmia and Mazury in Olsztyn (Poland). The species’ spawners were caught in Lake Dąbie in mid-May, i.e., during the natural spawning period. The fish were caught with fish traps and gillnets. They were carefully removed from the nets and put into containers with intensively aerated water [35].

The fish are very fidgety when being removed from the nets, which makes them susceptible to scale loss. This facilitates bacterial and fungal invasions, which can kill the fish within a short time. Therefore, all the fish handling was performed very gently. On the coast, the fish were put into transport bags with water and supplied with oxygen. The transport between the fish loading and release into the tanks lasted eight hours. Afterwards, the spawners were separated according to sex and put into two separate tanks working in one RAS system [36]. During fish collection in the wild, the water temperature was 14.0–14.5 °C.

It was easy to distinguish between the blue bream spawner sexes. The females’ scales during the spawning period are smooth or slightly rough. The abdominal section is swollen with a distinctly protruding papilla. The males’ snout, head and pectoral fins are yellowish and the bottom of their body is dark. There is a yellowish area of rough scales (easily felt with fingertips) between the anal fin and the caudal fin. Sexual dimorphism is poorly marked outside the reproductive period. The females’ readiness for reproduction was determined based on their appearance and by collecting oocyte samples according to the methodology described by Kujawa and Kucharczyk [37]. The oocytes were subsequently cleared in Serra’s liquid. They were in the third stage [17], which is the optimum stage for cyprinid fish stimulation for artificial reproduction [22,38]. Forty females and ten males were used in the experiment (Table 1).

The blue bream spawners’ body weight ranged from 404 to 995 g. The differences in the average body weight of females in different groups were relatively small.

The 1000 L tanks were made of plastic. They had independent systems for aeration, stepless regulation of the water temperature, EHEIM CLASSIC 2260 (EHEIM GmbH & Co.KG, Deizisau, Germany) external filters with a biological bed, EHEIM 1262 pumps with a flow of 3400 dm^3^/h and EHEIM (EHEIM GmbH & Co.KG, Deizisau, Germany) UV lamps for water treatment. Maintaining the set water temperature was ensured by a chilled water system and 2.5 kW heaters. Continuous water temperature monitoring and control were performed using the OXYGuard Pacific (OxyGuard International A/S, Farum, Denmark). The water temperature in the tanks was 14.5 °C (±0.1 °C) and it was not much different from that in Lake Dąbie at the time of fish catching.

Three preparations were used:


Ovopel, containing a mammalian GnRH analogue (D-Ala6, Pro9-Net-mGnRH) and metoclopramide as a dopamine receptor antagonist (Unic-trade, Budapest, Hungary) [29].CPH—carp pituitary homogenate (Argent, Redmont, WA, USA).Ovaprim, containing a salmon GnRH analogue (D-Arg6, Pro9-Net-sGnRH) and domperidone as a dopamine receptor antagonist (Syndel, Nanaimo, Canada) [30].


Only the females were stimulated as the males released sperm when their abdominal area near the sexual orifice was pressed slightly. The females were weighed on a KERN electronic balance with an accuracy of 0.1 g. Powdered carp pituitary homogenate (CPH) and Ovopel were pounded up in a mortar and dissolved in a 0.9% solution of NaCl. All hormonal injections were given intraperitoneally, under the left pelvic fin base. The volumes of the preparation doses are shown in Table 2.

The spawners in each experimental group were labeled with Floy Tags [39,40]. Females from all groups were kept together to minimize the impact of abiotic conditions on the final maturation process of females [11,13,15,20]. The control group included female fish receiving only physiological saline. The statistical analysis did not show any significant differences between females in the groups. After one day of acclimation, the water temperature in the tanks was increased to 16 °C (±0.1 °C) and the hormonal stimulation was started 24 h after the spawners were released to the tanks.

Later, the water temperature in the tank was increased to 19 °C (±0.1 °C) after the spawners were given the final (resolving) dose of the hormonal treatment. The spawner inspection was started 12 h after the final injection. After that time, the females were inspected every hour. All the fish handling was performed after they were anesthetized in an MS 222 solution (100 mg L^−3^ of water).

The eggs obtained from females were collected by gentle pressure on the abdominal wall from the head towards the tail and then placed separately into plastic vessels. Then, they were fertilized with semen from several males [34,41]. The same weight portions of eggs (10 g of eggs) from each female were fertilized with 100 µL of semen. The gametes were activated with water and then removed in the stickiness procedure before putting them into incubators. First, they were washed in a solution of urea with salt (40 of urea, 30 g of NaCl—10 L of water) and then for a few seconds in a tannin solution (7 g of tannin—10 L of water). Non-sticky eggs were placed in Weiss closed-circuit mini incubators (jar) with a capacity of 1.8 L [42]. For each of the females, such egg samples were produced in triplicate.

During the egg incubation, samples were taken randomly from each jar to determine the number of live eyed-egg embryos/total number of egg grains used for the observation ratio (%) and the number of swimming larvae/total number of egg grains used for the observation ratio (%).

The following parameters were also recorded during the experiment:


The weight of eggs obtained from each ovulating female—determined with an electronic balance with an accuracy of 0.1 g;The latency time—the time between the application of a stimulating agent and ovulation (h);The number of ovulating/stimulated females ratio (%);Spawner mortality rate in each experimental group (%).


The GSI was calculated in the following manner:GSI = 100 Wg × Wb **^−^**^1^
where: Wg—weight of the obtained eggs (g), Wb—total weight of the fish (g) before the eggs were collected.

The assessment of the normality of the data distribution, expressed as mean ±SD, was ascertained using Statistica software version 13.1 (StatSoft Inc. 2016, Tulsa, OK, USA). Data were normally distributed (Shapiro–Wilk test) and the variances were homogenous (Leven’s test). Differences between the means of the groups for each variable were evaluated using ANOVA and Duncan’s multiple range tests for group comparisons (*p* < 0.05 was considered to be significant).

## 3. Results

Gametes were not acquired from the fish in the control group. The lowest number of ovulating blue bream females (four fish) was found in group III, stimulated with Ovaprim (Table 3). There were six to nine ovulating females in the other groups, accounting for 60–90% of the females in the respective groups.

The latency time was the shortest in the spawners receiving Ovopel injections (group II). The fish in this group ovulated 12 h after the hormonal injection. Spawning was synchronized and all females spawned simultaneously (Table 3). The time between administration of the hormonal agent and ovulation was longer in the other groups, by 3–6 h, on average. The ovulation was considerably extended in time in group III, stimulated with Ovaprim. There was up to a seven-hour span between the ovulation of the first and the last female.

The fish survival rate was found to differ slightly depending on the hormonal preparations used to induce ovulation. The loss of females was the highest (30%—Table 3) among those stimulated with CPH. The survival rate was 90% in the group of spawners stimulated with Ovopel and 80% in the group in which Ovaprim was used.

Eggs with the highest average weight per fish (75.4 g) were obtained from the females in group II, stimulated with Ovopel (Table 4), while eggs with the lowest average weight per fish (49.5 g) originated from fish in group III. However, the differences were not statistically significant.

There was also no significant variation in the individual experimental groups in terms of the survival of the embryos to the eyed-egg stage (Table 4). The overall fertilization effectiveness expressed as the percent of live eyed-egg embryos was relatively high—from 73.8% in group III to 88.8% in group II, with the differences being statistically significant. The percent of swimming larvae was also high. It was the highest in group II (82.5%) and the lowest in group III (66.4%). The differences were statistically significant (Table 4).

## 4. Discussion

Hormonal stimulation is necessary to obtain gametes from most fish species living in the wild [43]. This especially applies to rheophilic cyprinids, whose reproduction under controlled conditions is often impossible without hormonal injections. Gametes can be obtained from them without such stimulation only when they are caught at a spawning site during the spawning [44]. Research into blue bream reproduction under controlled conditions following hormonal stimulation was based on studies of the bream *Abramis brama* [13,14] and rheophilic cyprinids [12,16,17,18,19,20]. Similarly to blue bream, these fish must be caught in the wild before their reproduction under controlled conditions. This results in high stress for the fish, with a possible decreased survival rate when they are put into tanks. Inducing ovulation in females and increasing the sperm amount by injection of CPH suspension has now been partly replaced with the use of other hormonal agents, e.g., Ovaprim or Ovopel [45,46,47,48,49,50]. Hormones are increasingly often combined with CPH injections. Developing a reproduction biotechnique involves testing various hormonal agents at different doses. It is of particular importance in cyprinids with diverse reproduction procedures. Fish producing eggs once per season must receive hormonal agents, often in combination with dopamine antagonists [45,46,50,51]. However, multiple-spawners should receive different hormones in different doses [13,52] than single-spawners [13,14,15]. Such stimulating agents include human chorionic gonadotropin (HCG), which is used as a supporting hormone in conventional methods of controlled reproduction. It is particularly recommended for use in percid reproduction [52,53]. This agent’s effectiveness in out-of-season reproduction of rheophilic cyprinids is very low. Moreover, the application procedure requires two to four injections, which is stressful for the fish and may result in their increased mortality [54,55]. According to Syndel, the Ovaprim manufacturer, the product shortens and synchronizes spawning, reduces fish stress, increases egg production, extends its activity and improves spawning effectiveness. However, the results are different in practice. Ovaprim use does not often give satisfactory results in reproduction, especially for cyprinid fish, particularly with respect to the percent of ovulating females and the biological quality of the eggs [56,57]. This was confirmed by the outcome of blue bream reproduction under controlled conditions in which both the number of ovulating fish and the egg weight were the lowest following the application of Ovaprim.

The hormonal agents used in this study included Ovaprim, Ovopel and carp pituitary homogenate (CPH). These agents are very often used in controlled reproduction of the carp and other cyprinids [22,25,58,59]. The ultimate effects of their use are similar, but their impact on the fish organism varies. CPH affects gamete maturation directly, whereas GnRH affects it indirectly—it influences the fish endocrine glands, which produce their own gonadotropin-releasing hormones and induce gamete maturation [17,25].

According to the study findings, ovulation in blue bream females caught in the wild can be stimulated with CPH, Ovopel or Ovaprim. Ovopel proved to be the most effective and Ovaprim the least. The use of Ovopel produced better results (number of ovulating females, spawning synchronization and the amount of eggs obtained) than the other reproduction stimulators. The survival rate for embryos in the eyed-egg stage for the blue bream females stimulated with Ovopel was higher than females receiving the other preparations. Importantly, the females in this group very well tolerated all reproduction-related handling under controlled conditions. Their survival rate until releasing the gametes was 90% (Table 3). Moreover, Ovopel is relatively cheap and easy to prepare [45].

CPH and Ovaprim were less effective than Ovopel. The female survival rate following stimulation with Ovopel was surprisingly high given the fact that they were acquired in the fishing process. CPH injection resulted in ovulation in 60% of the spawners (Table 2). The embryo survival rate in this group was only slightly lower than the rate for embryos hatched from the eggs from Ovopel-injected females. A similar observation was made for common dace *L. leuciscus* [11,20], asp *L. aspius* [12], ruffe *Gymnocephalus cernua* [31], common barbel *Barbus barbus* [32], river lamprey *Lampetra fluviatilis* [34], European catfish *S. glanis* [46], ide *L. idus* [48,54] and burbot *L. lota* [55]. The present study clearly shows that Ovopel is the best reproduction stimulator for blue bream caught in the wild. It proved to be highly effective and safe for the stimulated females. Their survival rate following Ovopel application was the same as in the control group. The Ovaprim effectiveness was low.

In many fish species, it has been found that the influence of abiotic factors such as dissolved oxygen (DO), pH, ammonium levels and temperature are important for the maturation of gonads and gametes. For example, even minimal temperature fluctuations in the range of optimal temperatures can have a very negative impact on the effectiveness of artificial reproduction [11]. For this reason, in our study, females from all groups were kept together to minimize the impact of factors other than the administered hormonal agents on reproductive efficiency [60]. Of course, when planning your research, it is important to take into account the issue of steroid hormones and the possibility of their excretion into the water by females. The most important is DHP (17,20β-dihydroxy-4-pregnen-3-one). However, for example, studies by Acharjee et al. on Asian stinging catfish (*Heteropneustes fossilis*) showed that the accumulation of DPH in the blood does not appear until many hours after ovulation [61]. Thus, the possibility of its excretion by some females and the impact on other breeders in our experiment was imperceptible, as evidenced by the results from the control group (ovulation—0%). Determination of the optimal stimulation conditions, including simultaneous environmental conditions and hormonal agents, is necessary for the effective artificial reproduction of cyprinids [32]. The studies conducted thus far on cyprinids have shown that off-season breeding, although more difficult to carry out, requires the use of thermal conditions that are the same as those used in the season [11,13,14,32]. The results of the reproduction of fish of the genus Abramis, which are related to the genus Ballerus, reported thus far have shown that the reproduction of fish in the season and in the off-season is very similar in terms of the reproductive protocol [62,63,64,65,66,67]. The research published by Kucharczyk et al. [13,14,15,65] showed that during off-season reproduction, the thermal conditions are identical to those during spawning in the breeding season, and only sometimes is the hormonal stimulation slightly different: slightly higher doses of hormonal agents or more injections are used. On this basis, it should be concluded that the given conditions of blue bream stimulation in the breeding season and outside the breeding season should be very similar.

The findings of this study confirm the effectiveness of Ovopel as a reproduction stimulator for fish caught in the natural environment. It is essential to have as many ovulating females as possible in the reproduction of wild fish under controlled conditions. Surviving females that fail to produce eggs are useless for further breeding. They also generate costs for their acquisition in the wild. Given the amount of work necessary to catch the spawners, the financial loss can be considerable. Currently, work is underway aimed at rearing blue bream spawners under controlled conditions. It is believed that partly domesticated fish will be less susceptible to stress and will better tolerate all reproduction-related handling [68].

## 5. Conclusions

Blue bream spawners were found to tolerate reproduction-related handling well under controlled conditions. Ovopel proved to be the best blue bream ovulation stimulator under such conditions. The rate of female survival until releasing the gametes following the use of Ovopel was the highest. The lowest female survival rate was observed following the use of CPH. High fertilization and larvae survival rates were observed in the groups of females stimulated with Ovopel or CPH.

## Figures and Tables

**Table 1 animals-11-01326-t001:** Characteristics of the examined blue bream *Ballerus ballerus* (L.)**.**

	Weight (g)	Length (cm)	Weight (g)	Length (cm)
	Females (*n* = 40)		Males (*n* = 10)	
min–max	404–995	34–48	298–443	33–39
mean	501.6	37.8	387.8	35.8
(±SD)	138.7	3.0	40.3	1.9

**Table 2 animals-11-01326-t002:** Treatments and their doses used for blue bream *Ballerus ballerus* (L.) female artificial reproduction.

Parameter	CPH (mg kg^−1^)	Ovopel (Pellets kg^−1^)	Ovaprim (mL kg^−1^)	0.9% NaCl (mL kg^−1^)
Group	I	II	III	Control
Injection	0.4 after 24 h 3.6	0.2 after 24 h 1.2	0.5	2

**Table 3 animals-11-01326-t003:** Effects of different hormonal agents on blue bream *Ballerus ballerus* (L.) female ovulation during controlled reproduction.

Group	I	II	III	Control
Number of stimulated females	10	10	10	10
Female body (min–max) (g)	404–964	416–995	475–846	462–836
Female survival after injection (%)	70	90	80	90
No. of ovulated females/(%)	6/60	9/90	4/40	-
Latency time (h)	15–18	12	13–20	-

**Table 4 animals-11-01326-t004:** Influence of different hormonal agents on egg quantity and quality during *Ballerus ballerus* (L.) artificial reproduction. The data in row with the same letter superscript are not differ statistically (*p* > 0.05).

Group	I	II	III	Control
	Mean (±SD)			
Female body (g ind^−1^)	494.5 (±22.4)	489.3 (±21.7)	467.4 (±48.0)	479.5 (±26.7)
Mass of eggs (g)	60.8 ^ab^ (±10.8)	75.4 ^a^ (±8.6)	49.5 ^b^ (±9.3)	0
GSI (%)	12.3 (±1.7)	15.4 (±2.1)	10.6 (±2.1)	
Survival to eyed-egg stage (%)	81.2 ^b^ (±0.6)	88.8 ^a^ (±0.5)	73.8 ^c^ (±1.2)	0
Swimming larvae (%)	74.3 ^b^ (±0.8)	82.5 ^a^ (±0.4)	66.4 ^c^ (±0.7)	0

## Data Availability

The data presented in this study are available on request from the corresponding author.

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
