# Peer review of "Artificial Reproduction of Blue Bream (Ballerus ballerus L.) as a Conservative Method under Controlled Conditions"

_animals, 2021, doi:10.3390/ani11051326_

Round 1

Reviewer 1 Report

General comment:

In the present study (Animals 1189992), the authors test different hormonal agents to produce viable gametes from blue bream, Ballerus ballerus. The study showed that broodstock tolerated handling procedures well and eggs were obtained from all hormonal treatments, while Ovopel proved to be the most promising hormonal agent for blue bream resulting in high fertilization and larval survival rates. Overall, the study is nice and has merit; however, parts of the manuscript need to be revised. In particular the Materials and Methods need to be restructured, rewritten and extended as important information on how the study was performed is lacking. More detailed comments are given below.

Specific comments:

Line 43: “representatives”

Line 50: “produces eggs” or “spawns”

Line 64: “reproductive functions”

Materials and Methods: More info needed on systems, tanks and fish rearing: E.g. type of system, water flow, RAS or flow-through, system size, tank size, density, light intensity how was temperature measured and how often (14.5 °C is not sufficient and I assume temperature was measured regularly which results in a value ± standard deviation). Please rewrite this part and substantively expand the information (not just limited to the above mentioned).

Materials and Methods: It is said that females were kept in two tanks. More details are needed on how females were kept during the hormonal treatment. I assume fish from different treatments were not kept in the same tank? Also, how were the eggs obtained, natural spawning of strip-spawning? Were males kept in the same tanks throughout the whole period. Currently, the Materials and Methods section is very confusing to read and needs to be substantively rewritten, restructured and extended.

Table 1: “length” (2 x); keep number of digits consistent and reasonable (the scale you mention has an accuracy of 0.1 g, the g value given in the table should then not exceed that accuracy), the length seems to have the wrong unit, I assume it is cm not mm?

Line 164: “were collected”

Line 169: “were placed”

Line 170: “egg incubation”

Line 185: “were collected”

Materials and Methods: How were the eggs incubated? More info on system, tanks and rearing methods is needed as for the broodstock above.

Materials and Methods: The description of the statistical analyses needs to be extended to match the standard for a scientific publication.

Lines 212-213: The sentence is confusing (and wrong) as you state afterwards there are significant differences between the treatments. Please rephrase.

Lines 222-230: This summary/interpretation does not belong into the results section.

Line 246: “produces eggs” or “spawns”

Line 255: “twice as long as”

Discussion: Long discussion on hCG considering that it has not been tested in the presented study.

Line 282: same as above “produces eggs” or “spawns”

General comment on references: The reference list should be kept a little more international, please redo some literature research and extend/replace references. E.g. I find it odd that a well-recognized and highly cited publication on broodstock management and hormonal manipulations in fish reproduction such as the one by Mylonas et al. 2010 is not cited once in this manuscript.

Author Response

Dear Reviewer,

Information about review was present in attached file.

Reviewer 2 Report

Dear Authors,

Introduction carefully written in accordance with the latest literature.
Materials and Methods
Statistics - write what tests were used.
The results are described concisely and accurately.
Tables: Ballerus ballerus write in italics
References
2. Brylińska is the newer edition from 2000, give different citation eg from Fishbase
9. Wrote no doi. Is he to be given?

Congratulations on the outcomes. The publication is written succinctly and honestly. It is of great importance for ichthyology and biodiversity protection.

ADDITION:

1. What is the main question addressed by the research?

The conducted research is an important methodological contribution to the reproductive process of the blue bream. The studies conducted so far do not fully answer the practical issues of the breeding ground. The research answers the important question of what is the best agent to stimulate female spiders for breeding purposes.

2. Do you consider the topic original or relevant in the field, and if so, why?

These studies are important due to the possibility of assisted reproduction. The blue bream is not an economically valued fish. However, it is an important species from the point of view of biodiversity. Perhaps in some ecosystems where it has become extinct it will be necessary to restock it.

New - there are no reports of artificial blue bream spawning.

3. Do you consider the topic original or relevant in the field, and if so, why?

These studies are important due to the possibility of assisted reproduction. The blue bream is not an economically valued fish. However, it is an important species from the point of view of biodiversity. Perhaps in some ecosystems where it has become extinct it will be necessary to restock it.

New - there are no reports of artificial blue bream spawning.

4. What does it add to the subject area compared with other published material?

The available literature does not accurately describe the course of artificial blue bream spawning. The general principles of restocking, including spreading, are presented (Marenko, Fedonenkov, 2016). The authors present in a concise and clear manner the execution of an artificial spout. And it is commendable.

5. What specific improvements could the authors consider regarding the methodology?

The methodology seems to be described sufficiently. The samples (10 females each) were good. The obtained effect confirms the effectiveness of Ovopel and CPH in the artificial reproduction of the blue bream.

Statistics - write what tests were used.

6. Are the conclusions consistent with the evidence and arguments presented and do they address the main question posed?

I believe that the conclusions are in line with the evidence and arguments. The publication provides the necessary information to conduct artificial blue bream spawning. It also contains information on difficulties that could affect the spawning process. This is a statement of practitioners that can also be used by people who deal with this subject.

7. Are the references appropriate?

The literature was chosen correctly. The authors cite the latest publications on the blue bream biology, reproductive cycle, as well as the methods of artificial spawning used in relation to other species.

  1. Brylińska is the newer edition from 2000, give different citation eg from Fishbase
    9. Wrote no doi. Is he to be given?

8. Please include any additional comments on the tables and figures.

Tables: Ballerus ballerus write in italics.

The tables are correct, they contain the most important information about the obtained results.

Marenkov, O.; Fedonenko, O. Ways of optimization of breeding conditions of fish by using artificial spawning grounds. Word Scientific News 2016, 49(1), 1-58.

Author Response

Dear Reviewer,
Information about present review is in attached file.

Round 2

Reviewer 1 Report

Table 1: Fix “Length” (spelled wrong twice)

Line 177: “in triplicates”

Materials and Methods: I do believe it is a flaw of the experimental set-up to keep females with different hormonal treatments in the same tank. How can you exclude that hormonal products are released into the water and affect the other fish? I would recommend changing this in future set-ups, fish with different hormonal treatments should be kept in separate tanks. Please check the literature whether anything is known on the release of the used hormonal products into the water. I assume it might be less drastic for the used products as it is for steroids, such as DHP. Please add a short paragraph reflecting on this issue in your discussion.

Author Response

Dear Editor,

The answers are included in the attachment.
